# Skin bacterial community dynamics of hands and forearms before and after military field exercise

Susanne Glenna,[1,2] Einar E. Birkeland,[1] Russell J.S. Orr,[2] Gregor D. Gilfillan,[3] Marianne Dalland,[3] Ole Andreas Økstad,[1] Øyvind A. Voie,[2] Trine B. Rounge[1,4]

**ABSTRACT** The human skin microbiome is crucial for health and immunity, especially under the extreme conditions military personnel face. Soldiers often encounter unique stressors and hygienic challenges that can alter their skin's microbial composition, particularly in field environments. In this study, we aimed to investigate the impact of military field exercises on the diversity and composition of the skin bacterial microbiota using 16S rRNA sequencing. We conducted a longitudinal study of Norwegian soldiers ($n = 19$) participating in outdoor training operations during the NATO winter exercise *Cold Response 2022*. Skin swabs were taken from soldiers' hands and forearms before and after the 10-day military exercise, and following a 3-week post-exercise leave. Our results reveal hand- and forearm-specific shifts in bacterial populations associated with the exercise, likely influenced by environmental exposure, reduced hygiene, and heightened social contact. Alpha diversity increased on forearms while remaining stable on hands, which appeared more resilient to perturbations. Both sites exhibited temporal changes in composition, with soil- and water-associated bacteria enriched post-exercise; most being transient on hands but more sustained on forearms. The soldiers' microbiomes converged during the exercise, then diverged in the post-exercise leave period, and neither skin site returned to baseline composition at follow-up. Our findings highlight the impact of collaborative outdoor activities on microbial communities and suggest that resilience and stability differ between skin sites.

**IMPORTANCE** Optimizing soldier health and resilience is critical for maintaining military readiness and operational effectiveness. The skin, as the body's first line of defense, is subjected to numerous challenges in military environments. Unique environmental and hygiene challenges can disrupt the skin microbiome and increase susceptibility to skin and soft tissue infections. This longitudinal research provides valuable insights into the effects of military service on the bacterial dynamics of the skin microbiome but can also inform hygiene management and disease prevention in comparable situations.

**KEYWORDS** skin microbiome, microbiota, 16S rRNA, bacteria, sequencing, metagenomics, military personnel, environment

T he human skin is home to a complex and dynamic community of microorganisms—including bacteria, archaea, fungi, and viruses—collectively known as the skin microbiome. This community plays a crucial role in maintaining skin health by reinforcing the skin's protective barrier against pathogens. The ecosystem of the skin microbiome is distinct across skin types—dry, moist, and sebaceous—each presenting a unique environment that shapes microbial residency (1). Regional differences in skin moisture, temperature, and pH drive microbial adaptation (2). Bacteria, the most abundant and well-characterized members of the skin microbiota, primarily belong to four main phyla: *Actinomycetota* (formerly *Actinobacteria*), *Bacillota* (formerly *Firmicutes*),

**Peer Reviewer** Bärbel Ulrike Foesel, Helmholtz Zentrum München, Neuherberg, Germany

Address correspondence to Trine B. Rounge, trinro@uio.no.

The authors declare no conflict of interest.

See the funding table on p. 13.

*Pseudomonadota* (formerly *Proteobacteria*), and *Bacteroidota* (formerly *Bacteroidetes*) (1, 3–5). Dry skin regions, such as the palm and forearm, are commonly the most diverse and least stable in community composition, while sebaceous sites typically have lower bacterial diversity dominated by lipophilic species like *Cutibacterium* (formerly *Propionibacterium*) *acnes* (*C. acnes*), likely due to high activity of sebaceous glands that may lead to a more exclusive and stable environment (1, 2, 4, 6).

The composition of the skin microbiome is shaped by host physiology, lifestyle, and environmental factors, including personal hygiene, physical exercise, climate, and geography (2, 7). Microbes can be transferred to the skin by contact with various surfaces, other individuals, pets, cosmetics, or the atmosphere (4, 8–10). Soldiers encounter additional, unique stressors that impact their skin's microbial makeup due to varying terrain, harsh weather conditions, cramped living conditions, and restricted access to hygiene amenities during combat, field exercises, and regular occupational settings. By altering the microbial balance, these extrinsic factors can affect skin health and susceptibility to infections (7, 11–14). The high prevalence of skin and soft tissue infections (SSTIs) among military trainees, due to close quarters and frequent physical contact, significantly impacts military readiness and economically burdens the Military Health System (15–18). This underscores the importance of understanding and mitigating impacts on the skin microbiome, as it plays a crucial role in preventing infections and preserving military personnel's health and readiness.

Research on the human microbiome has primarily focused on the gut, emphasizing the impact of lifestyle factors such as exercise (19, 20). By contrast, the skin microbiome—despite the skin's importance as the body's largest organ, dynamic interface, and protective barrier toward the external environment—has not been thoroughly characterized, especially under extreme conditions. This knowledge gap is particularly relevant for military personnel, who must maintain optimal health and performance under challenging conditions. While previous research has investigated the variability of the skin microbiome across time and body site in an individual and for certain dermatologic conditions (1, 4, 5, 21, 22), the soldier population provides a rare chance to explore how communal living and military field training impacts the bacterial community.

In this study, we aimed to investigate the impact of military field training on the skin bacterial microbiome of Norwegian soldiers. We analyzed 16S rRNA gene sequences from skin samples taken on the hands and forearms directly prior to and after a 10-day field exercise, as well as after a 3-week leave period. We hypothesized that the skin microbiome would undergo significant shifts during the field exercise due to increased environmental exposure and reduced hygiene, and thereafter gradually return to its original composition. Understanding skin microbiome changes can provide valuable insights into maintaining skin health in extreme conditions and inform strategies to mitigate potential negative effects of disrupted microbiomes in military and other high-stress environments. Our findings may contribute to the development of improved hygiene protocols and skin care strategies that support the health and well-being of soldiers and other individuals in demanding environments.

## MATERIALS AND METHODS

### Study design and participants

In this study, we recruited Norwegian soldiers participating in the outdoor NATO field exercise "*Cold Response*" during March 2022 to provide cutaneous swabs for DNA sequencing. Ethical approval for the study was granted by the Norwegian Regional Committee for Medical and Health Research Ethics in South-Eastern Norway (ref: 359040). Sample collection was performed after written, informed consent had been obtained from the subjects. Each participant was assigned a unique subject identifier number to use instead of their names to maintain their confidentiality and privacy. We used a longitudinal/repeated measures design where the same individuals were sampled at three different time points (sample rounds): (i) Baseline—the day before

the start of exercise, (ii) post-exercise—after the 10-day field exercise, and (iii) 3 weeks post-exercise (follow-up sampling after 3 weeks of military leave following the exercise; Fig. 1). All study participants ($n = 19$) were healthy Norwegian male professional soldiers, ranging in age from 20 to 30 years old (median age 23 years). Exclusion criteria included self-reported antibiotic treatment (oral or systemic) within 6 months before enrollment, observable dermatologic diseases or immunocompromised states, and other self-reported illnesses. Two soldiers in each of the second and third rounds of sampling were excluded due to ongoing COVID-19 illness, leaving 15 participants with samples collected at all three timepoints. Subjects were instructed not to shower, wash their hands, or use any skincare products for 12 hours prior to each sample collection time point. None of the subjects reported any issues with the swabbing procedure.

## Sample collection

Two samples (hypothenar palms (hands) and volar forearms (forearms)) were taken from each participant at three different sample rounds using skin swabs (Fig. 1). When applicable, an approximately $4 \times 4$ cm$^2$ area was sampled with a Becton-Dickinson Culture SwabsTM EZ Collection and Transport system (dual flocked swabs) soaked with a sterile wetting solution of 0.9% NaCl + 0.1% Tween-20 (Sigma-Aldrich). Swabbing was performed by three trained instructors according to a standardized protocol, at the same indoor location every time. For each participant, their left and right hypothenar palms (hands) were swabbed for 30 seconds each with the same dual-swab applicator, before immediately putting them back in their dry tube. This procedure was repeated for the volar forearms; left and right skin sites pooled together in a single sample. Unused swabs soaked in the sterile buffer and waved in the air for 60 seconds were used as negative controls for each sample round. The swabs were immediately put in the transport tubes and transported on dry ice until storage at −80°C.

## Total DNA extraction

All swabs were transported on dry ice (−78°C) to Clinical Microbiomics A/S (Copenhagen, Denmark) for high-throughput DNA extraction. Total DNA was extracted from the swab samples using the NucleoSpin 96 Soil (Macherey-Nagel) kit, with a low-input workflow, including up to 96 samples per extraction batch. For each sample collected, both swab heads in the sample tube were used to maximize yield. Chemical lysis (SL2 lysis buffer) and mechanical disruption of the cells (bead beating) were done for each swab, separately. Bead beating was done on a Vortex-Genie 2 horizontally at 2,700 rpm for $2 \times 5$ min. Then, the lysates from the swab pairs were combined, and the DNA was bound on a DNA-binding membrane, purified, and eluted according to the manual's instructions of the NucleoSpin 96 Soil kit. Negative controls (sterile water treated as a sample) and one positive control (ZymoBIOMICS Microbial Community Standard, Zymo Research, Cat No. D6300) were included in each batch. DNA concentrations were measured for all samples with the Qubit dsDNA HS assay (ThermoFisher Scientific), but only 13/102 samples had detectable DNA concentrations (>0.1 ng/μL). A randomly chosen subset of samples ($n = 22$) was measured with a more sensitive fluorometric approach for low biomass, using the AccuClear Ultra-High Sensitivity dsDNA Quantitation kit (working range 0.003–25 ng/μL), and the results are summarized in Fig. S1. The total amount of DNA in each skin sample was low, as expected, and all hand samples contained more DNA than forearm samples. All DNA samples were transported (on dry ice) to the Norwegian Sequencing Centre (NSC), Oslo, Norway for library preparation and sequencing.

## 16S rRNA amplification and sequencing

Amplification of the 16S V3-V4 (341F/785R) region was performed based on the two-step PCR procedure (with 28 + 8 cycles, see Fig. S2 for optimization tests) described in the Illumina application note (23), using primer sequences derived from Klindworth et al. (24). Human skin DNA (≤ 12.5 ng per sample), positive control (ZymoBIOMICS Microbial

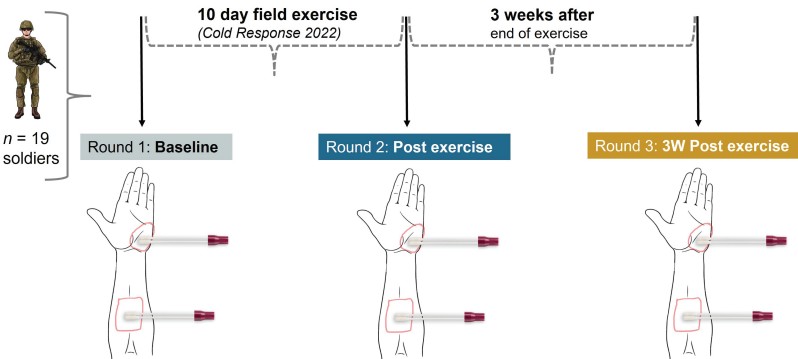

**FIG 1** Study design. Soldiers (*n* = 19) participating in a 10-day field exercise were swabbed on their hands and forearms for skin microbiome samples. Two samples (both hands and both forearms were pooled individually together) were taken for each subject in three rounds: (1) The day before the start of exercise (baseline), (2) immediately after the end of exercise, and (3) Three weeks (3W) after the end of exercise.

Community DNA Standard II (Log distribution), (Zymo Research, Irvine, CA, USA), Cat No. D6311), and negative control (sterile water) were used as templates, in addition to the air swabs and extraction controls mentioned above. The final libraries were verified using Tapestation 4200 with D1000 reagents (Agilent, Santa Clara, USA) and measured on a SpectraMax M3 fluorometric plate reader (Molecular Devices, San Jose, CA, USA) using Quant-it HS Assay reagents (Thermo Fisher Scientific, Waltham, MA, USA). Samples were pooled based on their concentrations and sequenced together on a MiSeq sequencing system (Illumina) with 300 bp paired-end reads (v3 reagents). 20% PhiX control library was added to the 16S libraries, and cluster density was reduced to 80% of regular levels. Base calling and production of demultiplexed fastq files were performed by running RTA v1.18.54.4 and bcl2fastq v2.20.0.422.

## Bioinformatics analyses

To ensure the high quality of the 16S amplicon sequencing reads, Fastqc v0.11.8 and multiqc v1.7 (25) analyses were performed both prior to and following the trimming of data. Adapters and low-quality sequences were removed from the raw reads using trimgalore v0.6.6 (26) with the parameters --length 150 -q 25. Primer sequences were removed using Cutadapt (v4.2) (27) with the following parameters: forward primer: CCTACGGGNGGCWGCAG, reverse primer: GACTACHVGGGTATCTAATCC, primer error 0.1 and primer overlap 3. Samples with <5,000 reads post-trimming were excluded from further analyses (*N* = 30, 6 biological +24 negative controls). Trimmed reads were imported into QIIME 2 v2024.2 (28) for taxonomic classification. Amplicon sequence variants (ASV) classification was conducted using the Divisive Amplicon Denoising Algorithm 2 (DADA2) plugin (29), including length trimming (p-trunc-len-f=270, p-trunc-len-r=210), denoising, merging, and chimera removal. These truncation parameters were chosen based on quality profiles from Qiime2 to ensure that only high-quality reads with sufficient overlap between read pairs (16 nts given the amplicon size used) were retained for classification. ASV classification was done using the SILVA 16S rRNA v138 database (30) with the qiime feature-classifier classify-sklearn (using a V3-V4 trained classifier) (31). ASV tables were filtered for mitochondria and chloroplasts, and singletons were removed. The ASV feature table, representative DNA sequences, taxonomy table, and sample metadata were imported from Qiime2 to R to create a phyloseq object for downstream analyses of microbiome data using the *phyloseq* (v1.42.0) R package (32). The taxonomic composition of positive control samples was compared to their expected theoretical distribution to investigate the accuracy of the classification.

Potential contamination during sampling, DNA extraction, and library preparation/sequencing was assessed using the *decontam* (v1.22.0) package (33). Only the negative controls remaining after the above-mentioned trimming, with their filtered ASVs, were used in the assessment of contaminants (9/33 samples: sampling [$n = 1$], DNA extraction [$n = 6$], and library preparation [$n = 2$]). Identification of contaminants was performed independently within each DNA extraction batch using the function *isContaminant* with the prevalence method at a threshold of 0.5 (features more abundant in negative controls are classified as contaminants). We observed an unimodal distribution of scores slightly above 0.5 (see Fig. S4A in the supplemental material), demonstrating that the chosen threshold was appropriate. The "decontaminated" data set with the identified contaminants removed was used in all downstream microbiome analyses.

## Statistical analysis

Statistical analyses were done in R version 4.4.1 (2024-06-14), using ggplot2 v3.5.0 for visualizations (34). Associations were considered significant at the $P < 0.05$ level, and all statistical comparisons were corrected for multiple testing using the Benjamini-Hochberg method unless stated otherwise. All samples were normalized to a sampling depth of 7,000 reads based on rarefaction without replacement (Fig. S5) before statistical comparisons (unless stated otherwise).

Microbial diversity was evaluated from the rarefied abundance of ASVs and analyzed individually for each skin site. Alpha diversity was assessed using richness (number of observed ASVs), Shannon, and Inverse Simpson indices to explore the within-sample diversity of microbial communities. The normality of the data was assessed using the Shapiro-Wilk test. Given that normality assumptions were not met for all indices, non-parametric tests were used for comparisons between sample rounds. The Wilcoxon signed-rank test was used to account for repeated measurements from the same individuals over time (pairwise differences). A linear mixed effects (lme) model was generated with *nlme* (v3.1–165) to adjust for batch effects and individual differences, using the formula *diversity ~sample round +batch, random = 1 + sample round | subject*, which includes both a random intercept and slope for the individual participant as a random effect. For all models, we performed *post hoc* pairwise comparisons between sample rounds using the *emmeans* package (v1.8.9) in R with the Tukey method for *P*-value adjustment.

Beta diversity was measured using the Bray-Curtis dissimilarity metric. Principal Coordinate Analysis (PCoA) was applied to visualize and interpret these dissimilarity patterns. To examine the impact of sample rounds on microbial community composition, a permutational multivariate analysis of variance (PERMANOVA) was conducted using *adonis2* from the R package *vegan* (v2.6–4) (35), with 999 permutations and estimation of marginal effects of terms (by="margin"). The analysis was complemented by an analysis of similarity (ANOSIM) when unequal variance between groups was identified through beta dispersion testing, using the same package. Beta dispersion was tested with permutest.betadisper in vegan with 999 permutations, followed by ANOVA and Tukey's honest significant differences (TukeyHSD) via the *stats* package (v4.4.1). Parwise Wilcoxon test with FDR correction was used to test for differences in Bray-Curtis distances between groups of samples. Differential abundance analysis was performed on non-rarefied reads, where only samples with sufficient sample depth (>7,000) were included. An additional filtering of features was performed to reduce the false discovery rate, where only features with a minimum skin-site specific frequency of 10 counts in at least 10% of samples were kept. The DeSeq2 package (v1.40.2) was used to perform differential abundance analyses with the negative binomial model-based Wald test implemented with sfType="poscounts" to account for sparsity (excessive zero counts), and *P*-values were FDR-adjusted to control for multiple testing (36). The design formula included subject-id and extraction batch. To generate more accurate log2 fold change estimates, the shrinkage estimator type="apeglm" was used with the function *lfcShrink* to shrink the resulting log2 fold change values toward zero (37).

## RESULTS

### Sequencing output and quality

In total, 7,420,302 raw paired-end reads were generated from all 141 samples, including 102 biological samples (51 hands, 51 forearms), 6 positive controls, and 33 negative controls. After removing samples with <5,000 trimmed reads, 96 biological samples (51 hands, 45 forearms) remained for further analyses, along with 6 positive and 9 negative controls. The total number of paired-end trimmed reads for the biological samples was 5,366,115 (median 52,511, range: 6,062–144,686). After quality-filtering, merging, and removal of chimeras, 2,666,595 high-quality reads were retained in the data set (median 26,167 per sample, range: 3,423–74,416). The number of reads remaining after each processing step is detailed in Table S1.

The taxonomic classification of positive control samples showed high consistency across technical replicates in both DNA extraction and sequencing controls (Fig. S3). We observed most of the expected genera, although some genera were over- or underrepresented in the DNA extraction controls, likely due to extraction bias in terms of varying efficiencies in lysing different bacterial cell walls (38). Some low-abundant taxa were not detected in sequencing controls due to the limitations in sensitivity. Negative control samples enabled the identification and removal of 49 potential contaminant ASVs in total (Fig. S4B). These represented 190,386 reads, accounting for ~7% of the (non-chimeric) reads overall (0.16%–1.54 % per biological sample). Table S2 lists all ASVs detected in the negative controls and whether they were classified as contaminants, along with their taxonomy and prevalence.

After trimming, feature-filtering, and decontamination, the total number of ASVs was 7180 (median per sample 292.5, range 4–1,071). Rarefaction to 7,000 reads kept all hand samples, but 15/45 forearm samples were excluded at this normalization step due to insufficient sample depth, leaving in total 81 samples left for 16S diversity analyses (Fig. S5). After rarefaction, the total number of ASVs was 6,863, with a median of 347 ASVs per sample (433 on hands and 160 on forearms, overall range of 53–905). Finally, as some batch effects remained after quality control (Fig. S6 and S7), although smaller than the study effects, the DNA extraction batch was included as a random factor to adjust for in statistical models.

### Characteristics of the skin microbiome of the hand and forearm in Norwegian soldiers

In total, the 6,863 ASVs identified were uniquely classified into 531 species, 682 genera, and 29 phyla (of which 24 phyla had >1 ASV). About 94% of all ASVs were classified to the genus level (of which ~10% were "uncultured"), while only 44% were classified to species level (of which ~65% were "uncultured"). Of the identified ASVs, 2,072 were shared between skin sites, while 4,266 and 525 were unique to hands and forearms, respectively. Both the bacterial diversity and taxonomic composition were significantly different when comparing the two skin sites at baseline. Alpha diversity (observed number of ASVs, Shannon and Inverse Simpson) was higher on hands than on forearms ($P < 0.001$, $P < 0.001$, $P = 0.0125$). There was also a significant difference in microbiome composition between hand and forearm samples ($P = 0.003$, PERMANOVA), and the proportion of community variance explained by skin site was $R^2 = 0.08313$ (Fig. S8). The three most abundant phyla on both skin sites were *Actinomycetota*, *Bacillota,* and *Pseudomonadota* (93% of total relative abundance), with the most abundant genera being *Staphylococcus* (16%), *Cutibacterium* (12%), and *Corynebacterium* (10%). Hands were also rich in *Massilia* (4%) and *Sphingomonas* (4%), while forearms had a relatively high fraction of *Lactobacillus* (4%) and *Micrococcus* (3%) (see taxonomic composition plot in Fig. 2). The rest of the genera—together accounting for 54% of all reads—were found at low abundance, <2.5% ("Other" in Fig. 2).

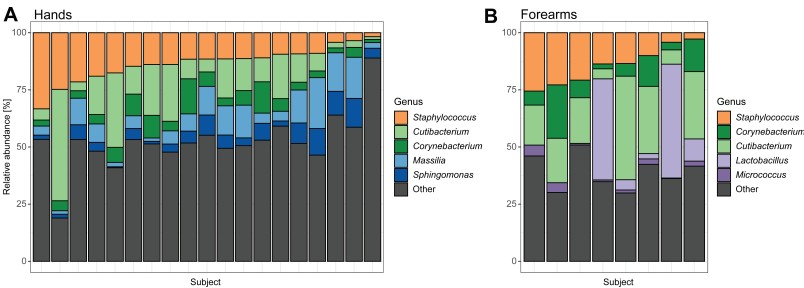

**FIG 2** Baseline taxonomic composition. Relative abundance (%) of top five genera (colors) in each subject at baseline (before the start of exercise) for (A) hand samples (*n* = 19) and (B) forearm samples (*n* = 8). Subjects are ordered by decreasing *Staphylococcus* abundance, the most dominant genus. Other: all other genera combined, each having less than 2.5% relative abundance.

## Temporal microbiota diversity

### Alpha diversity

The alpha diversity (within-sample diversity) on hands remained stable over time (Fig. 3A). No significant changes were found for either of the calculated diversity metrics (Shannon and Inverse Simpson), neither for unadjusted alpha diversity nor after controlling for repeated measures and batch effects ($P > 0.05$ for all comparisons; Table S3). However, when looking at the forearm samples, there was a significant increase in Shannon and Inverse Simpson diversity of forearm samples when going from baseline to post-exercise (Shannon: +35%, $P = 0.0024$ and Inverse Simpson: +128%, $P = 0.0445$, Fig. 3B). The overall change in alpha diversity during the study period (3W Post-exercise vs baseline) was also significant (Shannon: +42%, $P = 0.0002$ and Inverse Simpson: +302%, $P = 0.0034$). No statistically significant change was found between post-exercise and 3 weeks post-exercise for any of the diversity metrics. When comparing the richness (number of observed ASVs) of forearm samples, there was no significant difference between sample rounds.

### Beta diversity

To further assess differences in bacterial community composition, we used Bray-Curtis distances as a metric for beta diversity (between-sample diversity). Evaluating both skin sites and all time points together, subject identity (or inter-individual differences) accounted for the largest proportion (33.8%) of the observed variance in our model ($R^2 = 0.338$, $P = 0.001$, Table S4). Intriguingly, when comparing temporal composition, we found statistically significant changes ($P < 0.001$, ANOSIM) for both skin sites, demonstrated as clusters for sample rounds in the PCoA plot (Fig. 4), indicating a compositional shift caused by the field exercise. The composition differed significantly in both unadjusted and adjusted beta diversity. After adjusting for repeated measurements (subject IDs) and extraction batch, the sample round explained about 12% and 9% of the microbiome variability on hands and forearms, respectively (Table S5 and S6). The intra-individual variation is demonstrated in the plot by dashed lines connecting samples from each subject (Fig. 4).

Investigation of intra- vs inter-individual distances shows that there were no significant differences between the average intra-individual distance (variation within individuals over time) and the average inter-individual distance (variation between individuals at each time point) for either the hands or forearms ($P > 0.05$). This suggests that the compositional variation within individuals over time is comparable to the differences between individuals at any given time. While the magnitude of variation was similar between/within individuals, the stability differed between skin sites. The microbial composition on hands was more stable within individuals over time (i.e.,

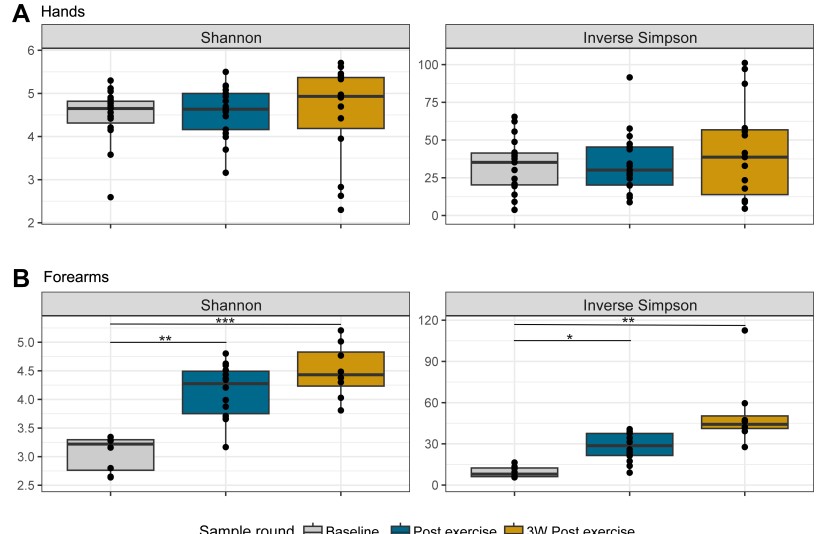

**FIG 3** Alpha diversity. Box plots of (unadjusted) bacterial evenness represented by Shannon and Inverse Simpson (InvSimpson) metric across the three sampling rounds for each skin site. (A) Hand samples (*n* = 51), (B) forearm samples (*n* = 30). Colors indicate sample rounds (gray = Baseline, blue = After exercise, gold = 3 weeks post-exercise). Boxes display interquartile range (IQR), the line is the median, and whiskers extend to 1.5 × IQR, with outliers shown beyond. Each black dot represents an individual subject. Statistical difference between rounds found by pairwise comparison of estimated marginal means after adjusting for the batch in a linear mixed model is marked by asterisks (significance levels: *: 0.05, **: 0.01, ***: 0.001).

had lower Bray-Curtis distances) than forearms (*P* < 0.001, Welch two-sample t-test), although some individuals varied more than others (Fig. 5A). Temporal stability within one skin site was not predictive of differences in the other, as evidenced by the lack of correlation between intra-individual Bray-Curtis distances of hand vs forearm samples (Pearson's product-moment correlation = 0.19, *P* = 0.4503). Furthermore, there was a non-significant trend toward increased similarity between hands and forearms within individuals during the exercise (*P* = 0.12), followed by a significant increase in the difference between the two sites in the 3 weeks post-exercise (*P* = 0.003, *emmeans*, Tukey method) (Fig. 5B).

To investigate whether soldiers became more similar to each other during the exercise, and if any changes persisted into the leave period, we analyzed the temporal dynamics of inter-individual Bray-Curtis distances and the beta dispersion for each skin site. Following the exercise, inter-individual distances significantly decreased on the hands (baseline: 0.63, Post-exercise: 0.59; *P* < 0.001, Wilcoxon signed rank), indicating community convergence (Fig. 5C). While a similar trend of decreasing distance was also observed on the forearms (median distance from 0.61 to 0.56), this change was not statistically significant (*P* = 0.18). Beta dispersion, reflecting individual microbiome spread, although lowest at Post-exercise, did not significantly decrease from baseline at either site (*P* > 0.05, Tukey's HSD). Interestingly, for both skin sites, inter-individual distances significantly increased in the 3 weeks post-exercise, with the follow-up round showing significantly higher distances than both baseline and post-exercise (*P* < 0.001 for all pairwise comparisons; hands: median = 0.75, forearms: median = 0.74) (Fig. 5C). The overall increase in beta dispersion was also significant (hands: *P* = 0.001, forearms: *P* = 0.003, ANOVA), with higher dispersivity in 3W post-exercise compared to the other sample rounds (*P* < 0.0015, Tukey's HSD) (Fig. 5D). Finally, microbiome composition at the follow-up sampling diverged significantly from baseline; Bray-Curtis distances between follow-up and baseline samples were significantly larger than those between follow-up

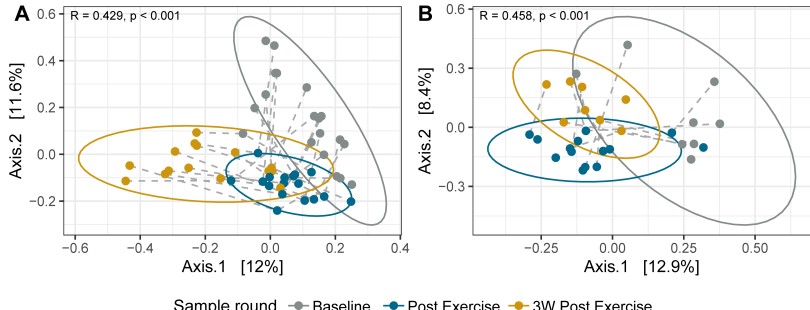

**FIG 4** Temporal beta diversity. PCoA plot of Bray-Curtis dissimilarity between the three sampling rounds for hand (A) and forearm (B) samples, with 95% confidence interval ellipses. Each dot is colored according to sample round (gray = Baseline, blue = After 10-day exercise, gold = 3 weeks after end of exercise), and dashed lines connecting samples from the same individual. The two axes show the principal components explaining the highest percentage of variances [denoted in brackets] in the communities. ANOSIM R values (range −1 to 1) indicate the magnitude of community variation attributable to sample rounds, with higher values reflecting greater differences *between* than *within* rounds. Significant differences between rounds were tested with ANOSIM (999 permutations).

and post-intervention ($P < 0.001$, Wilcoxon signed rank test), indicating a lack of return to the original state.

## Differentially abundant taxa

To investigate which taxa were significantly enriched or depleted, we tested for differential abundance using three contrasts: 2v1, post-exercise compared to baseline; 3v2, 3 weeks of "recovery" time; and 3v1, the total effect of exercise and 3 w break. At the chosen threshold (FDR-adjusted $P$-values $< 0.1$ and $|log2FC| > 0.2$), we found 138/725 evaluated ASVs to be differentially abundant (DA) in at least one contrast on hands, and 16/189 on forearms. Almost all DA ASVs (97%) were from the top three most dominant phyla (hypergeometric enrichment, $P < 0.001$). On hands, *Pseudomonadota* was the most prevalent, while on forearms, the DAs were similarly distributed between *Bacillota* and *Actinomycetota*. Figure 6 highlights the bacteria with the greatest relative changes in abundance in response to the exercise (contrast 2v1), while Table S7 provides the full DESeq2 DA results.

Focusing on the exercise intervention, we observed distinct patterns of change in the top differentially abundant ASVs on hands and forearms. On hands, while approximately half of the top 15 DA ASVs showed increased abundance, most of those that increased during exercise returned to baseline levels, while those that decreased in abundance remained low (Fig. 6A). By contrast, on forearms, the majority of the top DA ASVs exhibited sustained increases in abundance throughout the study period, with only one ASV showing a slight decrease in response to the exercise (Fig. 6B). During the 3-week leave period, a consistent trend of decreased abundance was observed across all top differentially abundant ASVs identified from the exercise intervention on both skin sites.

Indicating consistency across individuals, we found clustering of samples based on the abundance of the top 15 ASVs with the most substantial shifts on hands during the exercise, where there was a perfect separation between baseline- and postexercise samples (Fig. 6C). Interestingly, we also found that follow-up samples clustered more frequently with post-exercise samples ($P < 0.001$; binomial test). This clustering trend was not evident for forearm samples, where despite significant changes at the taxa level, there was no clear grouping of samples (Fig. 6D).

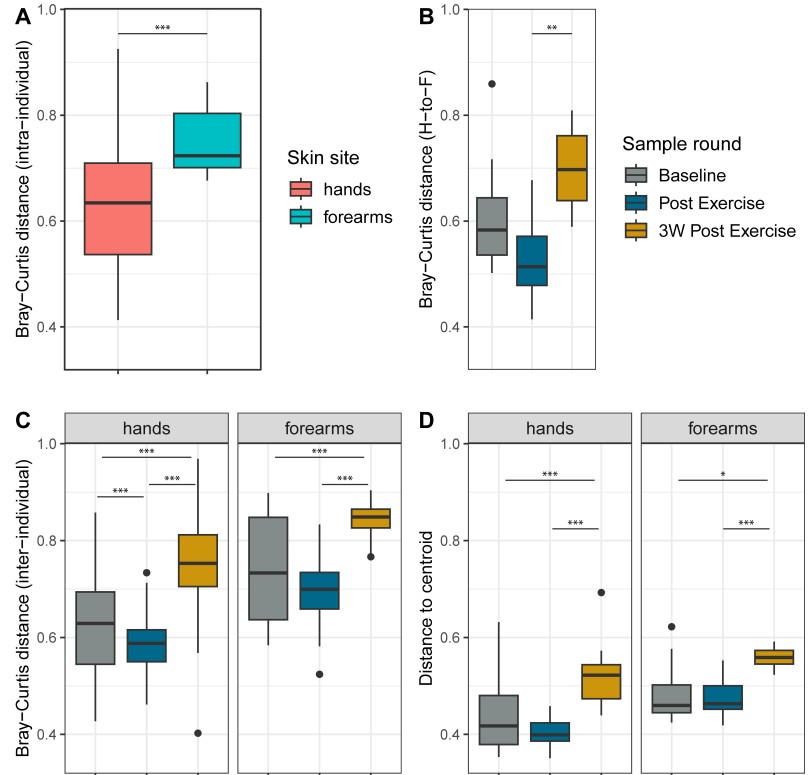

**FIG 5** Inter- and intra-individual variation. Box plots of distances *within* (A and B) and *between* (C and D) individuals over time. Boxes display interquartile range (IQR), the line is the median, and whiskers extend to 1.5 × IQR, with outliers shown beyond. Colors indicate sample rounds if not otherwise specified (gray = Baseline, blue = After exercise, gold = 3 weeks post-exercise). Statistical significance: *0.05, **0.01, ***0.001). (A) Bray-Curtis distance between samples from the same individual over time (intra-individual), within hands (pink) and forearms (blue). (B) Bray-Curtis distance between hands and forearms (**H-to-F**) within each individual for each round. (C) Bray-Curtis distances between individuals in each round. (D) Beta dispersion, distance to centroid for each individual in each round.

## DISCUSSION

Soldiers encounter environmental stressors and hygienic challenges that can impact their skin microbiome both during field exercises and in regular occupational settings. In this study of Norwegian soldiers, we observed distinct, skin site-specific shifts in bacterial composition after a 10-day winter exercise. Forearms showed increasing bacterial evenness, while the microbial diversity of hands appeared more resilient to the exercise intervention. Our study added further support to the skin microbiota being highly diverse and site-specific. Neither skin site returned fully to its original bacterial composition at follow-up.

The bacterial taxa identified in our skin samples were consistent with findings by others (1, 3, 4, 39). The dominant phyla on both hands and forearms were *Actinomycetota*, *Bacillota*, and *Pseudomonadota*, with *Staphylococcus*, *Cutibacterium*, and *Corynebacterium* as the most abundant genera, all commonly found on human skin. We observed high bacterial diversity on the soldiers' skin, in agreement with other skin microbiome studies (4, 39). The higher initial diversity on hands compared to forearms may result from regular contact of the hands with diverse surfaces, while forearms (often covered with clothes) may typically be less exposed to environmental microbes (39). The skin on the forearms might also be less hospitable to a wide range of microbes due to physiological factors such as hair follicles and sebum production, which are absent on palms (40, 41). It has been suggested that the volar forearm has a slightly more acidic (lower pH) environment than the hands, which tends to lower the diversity (42). Previous

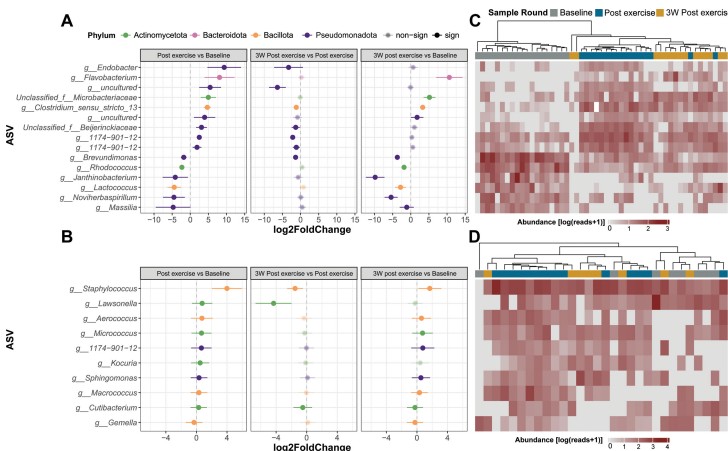

**FIG 6** Differentially abundant taxa. (A) and (B) Differential abundance comparing sample rounds for (A) hands (*n* = 51) and (B) forearms (*n* = 30). The top 15 differentially abundant ASVs for each skin site are shown, colored by phylum, and ordered by the effect size (log2 fold change ±SE) for the first contrast (post-exercise vs baseline). ASVs with absolute log2 fold change >0.2 and FDR < 0.1 are included. Opaque points indicate significantly different ASVs in each comparison. (C) and (D) Heatmaps of the abundance of ASVs (matched by row with A and B) with log10 +1 transformed counts (from low gray to high red) across samples (columns), hierarchically clustered (Bray-Curtis, Ward-D2) by sample round (gray = baseline, blue = post-exercise, gold = 3 weeks post-exercise) for (C) hands and (D) forearms.

research has suggested that skin sites with greater diversity tend to have lower temporal stability (1, 5, 43). By contrast, results from our study indicated that hands carried a more diverse and stable bacterial microbiome, both in terms of alpha and beta diversity. While there were no significant differences in alpha diversity on hands during or after the exercise intervention, we observed a significant increase in diversity (mainly evenness) in the forearm microbiome. On the hands, exercise induced both increased and decreased ASVs—resulting in changes to the overall composition (beta diversity) while maintaining a relatively consistent diversity within individual samples (alpha diversity). By contrast, the exercise primarily led to an increase in taxa abundances on the forearms, enhancing the evenness of the samples. The increased diversity during the outdoor winter exercise may reflect a combination of increased exposure to environmental bacteria from soil or snow, plants, and military equipment and physiological changes such as pH, moisture levels, sweat production, etc., associated with the activity. Previous studies have seen an increase in microbial diversity after exposure to soil and plant materials (44–46). While increased microbial diversity is generally considered beneficial for human health (46–48), further investigation is needed to assess whether such outdoor field exposure leads to functional improvements in host physiology.

As indicated by our analysis of beta diversity, the soldier population exhibited a shift in bacterial community composition after the exercise. This was true for both skin sites, although hands carried a more stable microbiome over time (Fig. 5A). Hands, being directly exposed to various surfaces and washed more frequently than forearms, may experience continuous microbial turnover that sustains a diverse, interconnected microbial network with enhanced redundancy and robustness to community disruptions (7, 49). Due to limited shower opportunities in the field, forearms might face a larger shift in hygienic conditions than hands, potentially leading to a larger influence of environmental microbes compared to usual conditions.

The differential abundance analysis suggests a shift influenced by environmental factors. Most of the differentially abundant taxa identified during the exercise are commonly found in environmental niches, especially in soil, plants, and water, reinforcing the idea that the outdoor environment was a major source of microbial input. The ASV with the highest increase in abundance due to the exercise was an uncultured bacterium from the poorly characterized genus *Endobacter*, of which the only known species has

been isolated from nodules in acidic soil (50). We also found a higher abundance of bacteria from the families *Microbacteriaceae* and *Beijerinckiaceae,* commonly involved in cycling processes of carbon and nitrogen in soils and aquatic environments (51, 52). *Flavobacterium urumquiense*, a psychrophilic species isolated from a glacier (53), was only observed in samples taken after the exercise, likely emerging due to the cold weather. It should be noted that several of the ASVs that decreased in abundance are also typically found in soil and water ecosystems, like *Brevundimonas*, *Janthinobacterium*, and *Massilia* (54–56). We also found environment-associated microbes increased on the forearms, such as *Sphingomonas*, *Aerococcus*, and *Kocuria marina* (57–59). However, normal skin commensals were also enriched, like *Staphylococcus*, *Micrococcus,* and *Cutibacterium*, potentially reflecting changes in skin oil production or moisture likely caused by exercise-induced sweating or environmental humidity (2, 60–62). As most of the differentially abundant taxa maintained their respective patterns of increase or decrease from post-exercise to follow-up, this suggests a prolonged response in the forearm microbiome. Interestingly, this sustained response contrasts with a previous study which found that most soil bacteria transferred onto forearm skin were no longer present after 2 hours (44). Most of the increased taxa found on hands were transient, decreasing in the following leave period. This suggests a varying degree of resilience across different skin sites.

Although there was a significant temporal variation, the largest difference in bacterial skin microbiome composition was identified between participants, in agreement with previous studies (3, 21). Intriguingly, we found that the soldiers' skin microbiome converged during the exercise, and then diverged in the post-exercise leave period. The convergence during field exercise may be due to increased social contact, for example when sharing tents and collaborating in certain tasks and activities. The soldiers also experienced shared environmental conditions (location, temperature, diet, etc.), potentially driving their microbiomes toward a more uniform state. A converging microbiome between individuals due to cohabitation or close-contact activities is a known dynamic for the skin microbiome (9, 10, 63).

Finally, the bacterial skin microbiome of the soldiers did not return to its original state after the exercise ceased. Instead, we observed a significant divergence between the follow-up and baseline microbiome composition when comparing the Bray-Curtis distances for each skin site. The lack of return to baseline emphasizes the potential for long-term shifts in composition, lasting at least 3 weeks after the intervention's conclusion. Although studies show that the human microbiome exhibits some resistance to change, its ability to recover after an alteration depends on the duration and nature of the exposure (5, 7, 64). Short-term exposures, like topical antiseptics usage, have been found to cause temporary and reversible shifts in microbial composition, indicating resilience to chemical disruptions (6, 65, 66). For example, it was demonstrated that most individuals' microbiomes recovered to near baseline composition within 2–4 weeks after chlorhexidine gluconate treatment (66). By contrast, more persistent disturbances, like systemic treatment with epidermal growth factor receptor inhibitors can lead to long-lasting dysbiosis (67). A longer recovery period beyond the 3 weeks examined in this study might have revealed a gradual return to baseline or stabilization of the microbiome at a new, alternative stable state (68, 69).

A notable strength of this study lies in the longitudinal design following the same soldiers before and after a large international military winter exercise. The study shows high technical robustness with trained instructors handling sample collection and using positive and negative controls to address the challenges of low microbial biomass and contamination. The limitations of the study include limited sample size and lack of a non-field exercise control group; however, the repeated measures design allows soldiers to serve as their own controls, thereby increasing statistical power. We also identified some batch effects introduced during DNA extraction, but these were smaller than the field exercise effects and adjusted for in our models. Despite the challenges with low

DNA input and loss of forearm samples, our sequencing data were of high quality, ensuring the reliability of our results.

Future studies may document hygienic practices and measure local environmental variables, such as temperature, humidity, and soil samples, to gain deeper insights into the observed findings.

## Conclusions

In conclusion, we have demonstrated changes in the skin microbiome on the hands and forearms of soldiers in response to an outdoor military exercise, identifying both spatial and temporal variability in the bacterial composition. The exercise was associated with an increase in environmental microbes typically found in soil, plant matter, and water. Our results enhance the understanding of the bacterial community in members of the armed forces, a group particularly vulnerable to infections. Further research is needed to understand the mechanisms driving the observed patterns of change within and between individuals and the potential impact of these changes on health outcomes. This knowledge could enhance soldier health, performance, and well-being in challenging environments, while also offering insights into infection control and hygiene practices beyond the military.

## ACKNOWLEDGMENTS

We want to thank Ragnhild Ueland at FFI for her contribution to the sample collection. We also acknowledge Clinical Microbiomics A/S (Copenhagen, Denmark) for performing high-throughput DNA extractions of all skin samples, and the Norwegian Sequencing Centre (Oslo, Norway) for library preparation and sequencing. The bioinformatics computations were performed on resources provided by Sigma2—the National Infrastructure for High-Performance Computing and Data Storage in Norway. We also acknowledge the assistance of OpenAI's ChatGPT (version 3.5 and 4.0) in providing programming syntax and linguistic suggestions for this paper.

## AUTHOR AFFILIATIONS

[1]Section for Pharmacology and Pharmaceutical Biosciences, Department of Pharmacy, University of Oslo, Oslo, Norway
[2]Norwegian Defense Research Establishment (FFI), Kjeller, Norway
[3]Department of Medical Genetics, Oslo University Hospital, University of Oslo, Oslo, Norway
[4]Department of Research, Cancer Registry of Norway, Norwegian Institute of Public Health, Oslo, Norway

## AUTHOR ORCIDs

Susanne Glenna ⓘ http://orcid.org/0000-0002-7730-0138
Einar E. Birkeland ⓘ http://orcid.org/0000-0002-9361-7987
Ole Andreas Økstad ⓘ http://orcid.org/0000-0001-9351-8535
Trine B. Rounge ⓘ http://orcid.org/0000-0003-2677-2722

## FUNDING

| Funder | Grant(s) | Author(s) |
| --- | --- | --- |
| Norges Forskningsråd | 332591 | Ole Andreas Økstad |
| | | Øyvind A. Voie |
| SoftOx Solutions | | Susanne Glenna |
| | | Øyvind A. Voie |

## AUTHOR CONTRIBUTIONS

Susanne Glenna, Conceptualization, Data curation, Formal analysis, Funding acquisition, Investigation, Project administration, Writing – original draft, Writing – review and editing | Einar E. Birkeland, Formal analysis, Investigation, Writing – review and editing | Gregor D. Gilfillan, Investigation, Writing – review and editing | Marianne Dalland, Investigation, Writing – review and editing | Ole Andreas Økstad, Funding acquisition, Project administration, Supervision, Writing – review and editing | Øyvind A. Voie, Conceptualization, Funding acquisition, Investigation, Project administration, Writing – review and editing | Trine B. Rounge, Investigation, Project administration, Supervision, Writing – review and editing.

## DATA AVAILABILITY

The sequencing data has been deposited in the NCBI Short Read Archive under BioProject ID PRJNA1159222. Code used to analyze the data can be found at: https://github.com/Rounge-lab/skin_microbiome_16S_seq_analyses.

## ETHICS APPROVAL

The study and experimental procedures were reviewed and approved by the Regional Committees for Medical and Health Research Ethics in South-Eastern Norway (ref: 359040, approved 03-02-2022). The participants provided their written informed consent to participate in this study.

## ADDITIONAL FILES

The following material is available online.

### Supplemental Material

**Supplemental figures and tables (Spectrum02953-24-s0001.pdf).** Figures S1 to S8; Tables S3 to S6.
**Table S1 (Spectrum02953-24-s0002.xlsx).** Metadata for 141 soldiers samples.
**Table S2 (Spectrum02953-24-s0003.csv).** ASVs detected in negative controls with contaminant classification.
**Table S7 (Spectrum02953-24-s0004.csv).** Differential abundance results.

### Open Peer Review

**PEER REVIEW HISTORY (review-history.pdf).** An accounting of the reviewer comments and feedback.

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
