## [Reviewer comments · Microbiology Spectrum]

Microbiology Spectrum

Skin bacterial community dynamics of hands and forearms before and after military field exercise

Susanne Glenna, Einar Birkeland, Russell Orr, Gregor Gilfillan, Marianne Dalland, Ole Andreas Økstad, Øyvind Voie, and Trine Rounge

Corresponding Author(s): Trine Rounge, Universitetet i Oslo

Review Timeline:

Submission Date:	November 18, 2024
Editorial Decision:	January 30, 2025
Revision Received:	March 12, 2025
Accepted:	March 17, 2025

Editor: Jan Claesen

Reviewer(s): Disclosure of reviewer identity is with reference to reviewer comments included in decision letter(s). The following individuals involved in review of your submission have agreed to reveal their identity: Bärbel Ulrike Foessel (Reviewer #2)

Transaction Report:

DOI: <https://doi.org/10.1128/spectrum.02953-24>

Re: Spectrum02953-24 (Skin bacterial community dynamics of hands and forearms before and after military field exercise)

Dear Prof. Trine B Rounge:

Thank you for the privilege of reviewing your work. Below you will find my comments, instructions from the Spectrum editorial office, and the reviewer comments.

Thanks for submitting your research to Spectrum. Your manuscript has now been evaluated by two independent Reviewers who are generally enthusiastic about your work (as am I). The Reviewers have brought up a couple of comments and suggestions to improve the manuscript and I would be happy to consider a revised version which addresses these in a point-by-point manner. Note that both Reviewers pointed out the relatively small sample size of the study, so make sure to include this potential limitation in the discussion of your paper.

Revision Guidelines

Sincerely,
Jan Claesen
Editor
Microbiology Spectrum

Reviewer #1 (Comments for the Author):

Article describes analysis of hands and forearm of 19 soldiers who underwent training, sampled before, after, and 3 weeks post. Methods for obtaining skin samples and performing 16S bacterial sequencing has multiple issues that could introduce artefacts but analysis is well performed.

Method for analyzing skin microbial composition is suboptimal, and likely even highly problematic. First - Meisel and Grice demonstrated in 2016 that 'skin microbiome surveys are strongly influence by experimental design' and specifically sequencing of V4 poorly captures skin commensal microbiota especially Cutibacterium PMID 26829039 ; second - 36 (28+8) cycles of PCR is too much as this is likely to highly enrich contaminants in the reagents and suggests that not enough microbiome DNA was in the clinical samples. 'Amplification of the 16S V3-V4 (341F/785R) region was performed based on the 2-step PCR procedure (with 28 and 8 cycles, respectively) described in the Illumina application note (23), using primer sequences derived from Klindworth et al. (24).' Is suboptimal. 9/33 negative controls reaching >5000 reads suggests amplification with 36 cycles reaches levels that call question on the dataset.

Method for sequencing is suboptimal. Most people now use staggered indexed barcodes. Authors state '20% PhiX control library was added to the 16S libraries, and cluster density was reduced to 80% of regular levels.' Reference : Fadrosch, D. W. et al. An improved dual-indexing approach for multiplexed 16S rRNA gene sequencing on the Illumina MiSeq platform. Microbiome 2,6 (2014).

A large number of the reads were lost. Why? Quality-filtering? Seems very concerning to lose 50% of your reads? This makes me worry about the dataset. 'The total number of paired-end trimmed reads for the biological samples was 5,366,115 (median 52,511, range: 6062-144,686). After quality-filtering, merging, and removal of chimeras, 2,666,595 high-quality reads were retained in the data set (median 26,167 per sample, range: 3423-74,416) (Table S1).

This makes me very concerned. Some contaminants will be in the reagents but others could be skin microbes from another source (practitioner obtaining sample, skin shedding in the air). Need more specifics about how negative controls were used and what was removed from dataset. 'Negative controls enabled the identification and removal of 49 potential contaminant ASVs in total (Fig. S3B).'

Sphingomonas is a common water contaminant. Was this explored? 'Hands were also rich in Sphingomonas (4 %),'

analysis was nicely presented.

Reviewer #2 (Comments for the Author):

The manuscript submitted by Glenna et al. investigates the impact of stressors, in this case given by a military field exercise on the human skin microbiome. It is a very well conducted and described study with a clear - although not unexpected - outcome. My major point of concern is the small sample size of only 19 individuals initially with a significant loss of samples especially from the forearm group, I think.

Moreover, one formality I'm missing is line numbers. It would be very helpful for the review process, if one could just give the line number to address specific issues!

The rest is minor comments:

Keywords: I don't fully understand why "soil" is listed and would suggest, to rather add something like "military field exercise" and/or "field conditions" to address the stressors investigated.

p.3, introduction, first lines: Explicitly name "barrier function"?

p.5, 2.1 study design and participants:

"informed consent had been contained" instead of "was obtained"?

What is "anonymous identification numbers"? - Rephrase?

p.6, last sentence before the figure:

"12 hours prior to" instead of "before"?

Caption, figure 1 & p.7, 2.2 versus p.7 2.3:

In the figure caption and sample collection part it looks like/sounds like the left and right palm and forearm would have been swabbed with the same swap already. However, in the description of the DNA extraction procedure it says "for each sample, both swabs were used to maximize yield". - This is a bit confusing. - Please clarify!

p.6, 2.2:

Which area (approximate/defined size?) has been swabbed?

p.13, 3.2, first sentence:

Change "belonged to" to "were assigned to".

Moreover, I wonder how it can be more genera than species? I would expect it to be the other way round ... If it's unclassified units on species level that make this discrepancy, they should be counted as well (maybe separately?) to explain this, I think.

Figure 4:

It is not really visible which points are interconnected by the dashed lines. - Maybe think of something else as e.g. an additional panel with a different colour code for the individuals instead of the timepoints.

p.21, 3.4, towards the end of the first paragraph:

Isn't it figure 6 (instead of 5) the text refers to?

p. 24, first sentence:

1) I don't think that "microbial turnover" is the right expression in this context, 2) I moreover think that "stable" isn't the right expression to describe the alpha diversity. - Please revise/rephrase.

p.24, relatively to the end:

I don't fully understand & I'm not sure, if the statement is true that "forearms are more exposed to bacteria"? - Do you mean from outside, as those aren't removed that often? - If so, on the other hand the normal skin flora also isn't removed/affected by washing that often and this probably would protect against "invaders" from outside ... I'm not sure, if I'm thinking along the same lines as you making this statement. However, please consider and rethink/rephrase if necessary.

p.24, somewhere in the middle:

I wonder, if it's just "normal skin commensals" that are enriched, when it comes e.g. to Staphylococcus. - If not already done, maybe you should examine your data set for the raise of potentially harmful species linked to e.g. eczema as an additional point of discussion.

Point-by-point response, Glenna et al, “Skin bacterial community dynamics of hands and forearms before and after military field exercise”

Dear reviewer #1,

Thank you for your constructive feedback on the methodology in this study. We appreciate your concerns regarding the sequencing, quality-filtering, and handling of negative controls, and address these point-by-point below. We must emphasize that our methodological choices were limited by the low-biomass nature of these skin samples.

- **Concern #1:** Sequencing of V4 poorly captures skin commensal microbiota especially *Cutibacterium*

Response: We agree that the choice of primers is important to optimize coverage and reduce bias in microbiome studies, and support Zeuween et al. in their proposed need for standardized skin microbiome-specific primer sets (1). We want to emphasize that we employed the V3-V4 hypervariable regions for sequencing, not just V4 alone. Although several studies have commented on the inherent biases in this region and the V1-V3 region (2-4), the V3-V4 region is widely used in skin microbiome studies (5-8) and has been shown to achieve the closest sequence match to known low-diversity mock community samples (9). Additionally, the sequencing was conducted through a sequencing center that incorporates the V3-V4 primers into their established pipeline. A different approach would have required significant modifications to the center's validated workflow, potentially impacting project timelines and quality. Nevertheless, all samples are processed consistently and simultaneously to ensure comparability between groups. Finally, we did identify *Cutibacterium* in our samples; it was in fact one of the top three most abundant genera (see Figure 2).

- **Concern #2:** [...36 (28+8) cycles of PCR is too much as this is likely to highly enrich contaminants in the reagents and suggests that not enough microbiome DNA was in the clinical samples.]

Response: These samples contain low biomass, and therefore require extensive amplification. We optimised the 2-step amplification accordingly to use the minimum number of cycles possible to obtain sufficient material to sequence. 25/30/35 cycles of PCR 1 were tested. We decided to go for 28 cycles as several samples were too low to sequence for 25 cycles, while the negative control was barely detectable at 30 cycles (see figure below). We agree that less amplification would be desirable, but without this number of cycles there would be no dataset. We have now added these optimisation steps in Supplementary figure S2.

- **Concern #3:** 9/33 negative controls reaching >5000 reads suggests amplification with 36 cycles reaches levels that call question on the dataset

Response: While a 30% positive rate for negative controls may seem like a cause for concern, the absolute numbers of negative control reads obtained is not directly comparable to the study samples. Control amplification often results in sub- or on-the-limits of detection amounts of DNA, so it is impossible to combine with other sample libraries on an equimolar basis. Rather, "as much as you can" is added to a pool. This does not allow for a quantitative deduction of contaminant reads, but rather a qualitative exclusion of species based on prevalence in control samples, as we have done (see comments regarding the handling of negative controls; concern #6).

- **Concern #4:** Why not use staggered indexed barcodes?

Response: We agree that staggered primers are far superior and routinely use the method the reviewer refers to as a first choice. However, the full-length primers employed in the Fadrosch et al method do not work well on low-biomass samples, and perform best on bacteria-rich samples such as fecal or soil samples. It appears that full length primers do not amplify as efficiently as shorter ones, which is why the two-step amplification was chosen (and was in fact necessary) for this project.

- **Concern #5:** Why were so many reads lost during preprocessing, seems concerning?

Response: While substantial read loss during pre-processing may be a concern, we made a choice to focus this ASV-based analysis on high-quality reads, as drop-offs in quality can have large impacts on detected variants. It is also not surprising to lose considerable amounts of reads when starting from such low biomass samples and sequencing quite deep. Many reads are likely dimers. Our rarefaction curves (Fig. S4) demonstrate that the remaining reads are sufficient to capture the majority of the diversity in the samples, making the loss of reads less concerning.

Most of the lost reads were filtered out in dada2 due to being too short after quality filtering and removal of adapters and primers. On average, 58% (SD 3) of the sequences passed the length filtering step. Losing about 40 % of reads at this filtering step is expected and not a major cause of concern (10). We assessed the quality profiles from Qiime2 (using `qiime demux summarize`) to determine appropriate truncation points where the median quality score begins to decline. After testing

different filtering parameters, we decided on p-trunc-len-f=270 and p-trunc-len-r=210 as the best option to retain most reads of high quality while ensuring sufficient overlap between forward and reverse reads. These parameters result in a sequence length of 480 nts, and since the amplicon size was 464 nts, we had 16 nts overlap (at least 12 nts overlap is required). The merging of forward/reverse reads was successful (just 4 % reads lost on average), and there were not many chimeric reads (mean=964, median=195, in total 1.7 % of trimmed reads). We kindly refer to Table S1 for more details of the number of reads left at each step of the process (and mention this explicitly in lines 268-269). We have also added some more details in the methods section (see lines 195-198).

- **Concern #6:** Need more specifics about how negative controls were used and what was removed from dataset

Response: As stated in our manuscript (see lines 277-278), we identified 49 ASVs in the negative controls that were more prevalent in negatives than in true samples. These were classified as contaminants using *decontam()*, and removed from biological samples containing those sequences - see also response to concern 3 above; we consider this filtering of potential contaminants to be relatively conservative. We agree that more information about what was removed is beneficial. We have therefore included a supplementary table (Table S2) listing all 79 sequences found in negative controls and whether they were identified as contaminants (referred to in lines 279-281). This table shows their prevalence in both skin samples and each type of negative control, making it possible to discern at which processing step any potential contamination has occurred.

- **Concern #7:** *Sphingomonas* is a common water contaminant. Was this explored? 'Hands were also rich in *Sphingomonas* (4 %)'.

Response: Out of the 49 ASVs identified as potential contaminants from the negative controls, only 1 of them was classified as *Sphingomonas*. This ASV was only found in one negative sample (sequencing control) with 26 reads and was absent in all biological samples, making it highly unlikely that the *Sphingomonas* taxa observed on the hands were caused by technical water contamination.

- **Additional change:** We have also made a change to the sentence in lines 272-275, as it is more likely that extraction bias explains the over/underrepresentation since the difference to theoretical in relative abundance was seen only for extraction controls, not sequencing controls. Gram-negative species being overrepresented and Gram-positive underrepresented is a recognized phenomenon (11).

Dear reviewer #2,

Thank you very much for your thorough review and valuable feedback on our paper. We appreciate your concerns regarding the sample size and the loss of samples, as well as your comments on specific sentences. We kindly refer you to the marked-up manuscript for review of adjustments we have made based on your feedback.

- **Concern #1:** Small sample size (19 individuals initially)

Response: We appreciate your concern regarding the sample size and would like to address this issue. Since our study focuses on military personnel, the population presents inherent limitations in terms of accessibility; logistical and operational constraints associated with working with this unique group constrain the feasible number of participants. While we acknowledge that a larger sample size would provide increased statistical power and enhance the reliability of our findings, we believe our study design contributes to increased robustness given the limited sample size. Our study employs a repeated measures (within-subjects) design, where each participant serves as their own control. This design significantly reduces inter-individual variability and increases statistical power, enabling meaningful comparisons even with a relatively small sample size. According to statistical literature (e.g. Field, 2018; Cohen, 1988), within-subject designs require fewer participants than between-subject designs due to lower within-group variability. However, as we consider this concern to be valid, we have included the small sample size as a limitation in the discussion (lines 541-544).

- **Concern #2:** Significant loss of samples, especially from the forearm group

Response: Skin samples are known to have a low microbial biomass, which makes them challenging to work with. Loss of forearm samples was mainly due to insufficient sequencing depth resulting from the very low DNA amount (see Supplementary Fig. S1). During trimming of raw reads, we decided on a threshold of 5000 trimmed reads for further processing to ensure adequate data quality and reliable representation of microbial communities in each sample. This threshold resulted in removal of 6/51 forearm samples, along with most negative controls. After dada2-filtering, we chose to normalize all samples to 7000 reads based on rarefaction curves, keeping most of the diversity of the data (see Supplementary Fig. S4). This step excluded another 15 forearm samples, but these would have been lost even with rarefying to 5000 reads (we chose the highest sample depth that would keep the same amount of samples overall). We have mentioned this limitation in the discussion section (lines 544-547).

- **Concern #3:** Missing line numbers

Response: The journal checklist stated that continuous line numbers were not necessary for the first submission and would be automatically added to the Word-file during revision. However, we have now added line numbers to the marked-up-manuscript ourselves, so I hope that it makes it easier for the rest of the review process.

We also address some of your minor comments below, but refer you to the revised manuscript for the linguistic changes:

- **Comment #1:** Keywords: Why was "soil" listed? I would rather suggest adding something like "military field exercise" and/or "field conditions" to address the stressors investigated.

Response: "Military field exercise" specifically was not chosen as it is part of the title, which is already being indexed by search engines, so we prioritized unique new keywords to increase the paper's visibility to a diverse audience. "Soil" was previously chosen as it was one of the most common environmental exposures we observed, and it is a MESH term (controlled vocabulary thesaurus for PubMed). We have replaced "soil" with "environment" in the revised manuscript.

- **Comment #2:** Explicitly name "**barrier function**" in the first sentence of Introduction (lines 55-58)?

Response: Yes, we have now added it there explicitly, although the barrier function was mentioned later in the Introduction (see lines 84-85).

- **Comment #3:** Line 110: "informed consent had been contained" instead of "was obtained"?

Response: The phrase "had been *contained*" is not applicable in this context, as it refers to something being enclosed or held within limits, which does not pertain to the process of obtaining/acquiring consent. So in the revised manuscript we state "informed consent had been obtained".

- **Comment #4:** Rephrase "**anonymous identification numbers**"?

Response: Suggested clarification in lines 110-112: "Each participant was assigned a unique subject identifier number to use instead of their names to maintain their confidentiality and privacy".

- **Comment #5:** Line 123: "**12 hours prior to**" instead of "before"?

Response: Accepted change.

- **Comment #6:** Clarify swabbing procedure: In Fig. 1 caption and section 2.2 it looks/sounds like the left and right palm and forearm would have been swabbed with the same swab already. However, in the DNA extraction description (section 2.3) it says "for each sample, both swabs were used to maximize yield".

Response: Thank you for bringing this to our attention. We acknowledge that the wording may cause some confusion. To clarify, each sample was collected using a dual swab system (as mentioned in section 2.2). This means that for every sample, we used a single tube containing two swab heads connected to the same cap. This system allows us to use both swab heads for one sample to maximize DNA yield, as described in the DNA extraction procedure. Our initially submitted version of figure 1 included a barely visible depiction showing that each swab system has double swabs, but we have updated Fig. 1 to clarify this for the revision. We hope this explanation resolves any confusion, and refer you to the tracked changes in section 2.2 and 2.3 for a clarification in the revised manuscript.

- **Comment #7:** Which area has been swabbed?

Response: In section 2.2 we described the area as approximately 4x4 cm² (see lines

135-136), but to be precise this was the size for the volar forearm area, while the hypothenar palm is already a predefined area in the literature. Below we have added a sketch of the swabbed areas, and this has been included in the revised Fig. 1 for added detail.

- **Comment #8:** Line 291: Change "belonged to" to "were assigned to".
Response: We have changed it to "were uniquely classified into".
- **Comment #9:** How can there be more genera than species? I would expect it to be the other way round ... If it's unclassified units on species level that make this discrepancy, they should be counted as well.
Response: This is due to the resolution of 16S rRNA amplicon sequencing, which for most ASVs cannot distinguish between species and therefore only classifies down to genus-level. To clarify, the numbers we reported in our manuscript were the number of *different* uniquely classified units at each taxonomic level. For better transparency of the taxonomic classification, we have now added the proportion of unclassified units in our revised manuscript (lines 292-294).
- **Comment #10:** Figure 4: It is not really visible which points are interconnected by the dashed lines. Maybe think of something else as e.g. an additional panel with a different colour code for the individuals instead of the timepoints.
Response: Thank you for your thoughtful suggestion regarding Fig. 4. The purpose with the dashed lines is to provide an overall perspective on intra-individual variation, rather than to trace individual connections. We recognize that this choice may make it difficult to discern specific linkages between points, and we had considered this concern during the figure's design. However, we believe that the current format effectively highlights the overall trends, with a main focus on the sample round effect. We appreciate your understanding and hope this explanation clarifies our intent.
- **Comment #11:** line 417: Isn't it figure 6 (instead of 5) the text refers to ?
Response: Yes, you are correct. The reference should indeed be to Figure 6, not Figure 5. The confusion arose because we originally had an error where two figures were mistakenly both labeled as "Figure 2." Thank you for bringing this to our

attention. We have thoroughly reviewed the manuscript and have corrected all figure numbering errors to ensure consistency throughout the revised manuscript.

- **Comment #12:** lines 468-473: 1) I don't think that "microbial turnover" is the right expression in this context, 2) I moreover think that "stable" isn't the right expression to describe the alpha diversity. Please revise/rephrase.

Response: Thank you for bringing the terminology to our attention. We have used quite some time to phrase that sentence, but we agree with you that it might not use the best words to describe what we mean. We wanted to highlight that one explanation for no significant change in alpha diversity on hands could be due to the "exchange" (previously named turnover) of certain ASVs, since there was a similar number of increasing and decreasing ASVs on hands. But we acknowledge that "microbial turnover" typically implies a more extensive replacement or cycling of species within microbial communities. In the context of our study, which focuses on the balanced alteration in the abundances of specific ASVs rather than a wholesale replacement of the community, this term may not accurately convey our findings. We have reworded accordingly as follows (see lines 468-473): "On the hands, exercise induced both increased and decreased ASVs - resulting in changes to the overall composition (beta diversity) while maintaining a relatively consistent stable diversity within individual samples (alpha diversity)".

- **Comment #13:** lines 487-489: What do you mean by "'forearms are more exposed to bacteria"?

Response: Thank you for pointing out the ambiguity in the original sentence. Our intention was to convey that, due to limited shower opportunities in the field, the forearms may experience less frequent washing than usual, leading to a more pronounced change in hygiene routines for the forearms compared to the hands during the exercise. We have amended our wording accordingly (lines 487-489).

- **Comment #14:** lines 504-507: I wonder if it's just "normal skin commensals" that are enriched, when it comes e.g. to *Staphylococcus*. If not already done, maybe you should examine your data set for the rise of potentially harmful species linked to e.g. eczema as an additional point of discussion.

Response: Thank you for your valuable suggestion. We acknowledge the importance of understanding the potential implications of specific bacterial populations, such as *Staphylococcus*, in the context of skin conditions like eczema. However, as our study utilized 16S rRNA sequencing, the taxonomic resolution is limited to higher taxonomic levels, typically up to the genus level, and does not allow for reliable differentiation between species or strains. As a result, while our data can indicate shifts in the abundance of genera such as *Staphylococcus*, it does not provide the resolution necessary to identify specific pathogenic or associated strains linked to conditions like eczema.

References

1. Zeeuwen PLJM, Boekhorst J, Ederveen THA, Kleerebezem M, Schalkwijk J, van Hijum SAFT, Timmerman HM. 2017. Reply to Meisel et al. *J Invest Dermatol* 137:961–962. <https://doi.org/10.1016/j.jid.2016.11.013>.
2. Meisel JS, Hannigan GD, Tyldsley AS, SanMiguel AJ, Hodkinson BP, Zheng Q, Grice EA. 2016. Skin microbiome surveys are strongly influenced by experimental design. *J Invest Dermatol* 136:947–956. <https://doi.org/10.1016/j.jid.2016.01.016>.
3. Conlan S, Kong HH, Segre JA. 2012. Species-level analysis of DNA sequence data from the NIH Human Microbiome Project. *PLoS One* 7:e47075. <https://doi.org/10.1371/journal.pone.0047075>.
4. Kong HH. 2016. Details matter: designing skin microbiome studies. *J Invest Dermatol* 136:900–902. <https://doi.org/10.1016/j.jid.2016.03.004>.
5. Zeeuwen PLJM, Boekhorst J, van den Bogaard EH, de Koning HD, van de Kerkhof PMC, Saulnier DM, van Swam II, van Hijum SAFT, Kleerebezem M, Schalkwijk J, Timmerman HM. 2012. Microbiome dynamics of human epidermis following skin barrier disruption. *Genome Biol* 13:R101. <https://doi.org/10.1186/gb-2012-13-11-r101>.
6. Prast-Nielsen S, Tobin A -M., Adamzik K, Powles A, Hugerth LW, Sweeney C, et al. Investigation of the skin microbiome: swabs vs. biopsies. *British Journal of Dermatology* 2019;181:572–9. <https://doi.org/10.1111/bjd.17691>.
7. Kim, JH., Son, SM., Park, H. et al. Taxonomic profiling of skin microbiome and correlation with clinical skin parameters in healthy Koreans. *Sci Rep* 11, 16269 (2021). <https://doi.org/10.1038/s41598-021-95734-9>
8. Nørreslet LB, Lilje B, Ingham AC, Edslev SM, Clausen ML, Plum F, Andersen PS, Agner T. Skin Microbiome in Patients with Hand Eczema and Healthy Controls: A Three-week Prospective Study. *Acta Derm Venereol.* **2022** Jan 18;102:adv00633. doi: 10.2340/actadv.v101.845. PMID: 34877605; PMCID: PMC9631265.
9. Castelino M, Eyre S, Moat J, Fox G, Martin P, Ho P, Upton M, Barton A. 2017. Optimisation of methods for bacterial skin microbiome investigation: primer selection and comparison of the 454 versus MiSeq platform. *BMC Microbiol* 17:23. <https://doi.org/10.1186/s12866-017-0927-4>.
10. Callahan, Benjamin. "Loosing 50-75% of sequences after filtering and denoising." *benjjneb/dada2* Issue #1164 (2022). <https://github.com/benjjneb/dada2/issues/1164>.
11. Elie C, Perret M, Hage H, Sentausa E, Hesketh A, Louis K, et al. Comparison of DNA extraction methods for 16S rRNA gene sequencing in the analysis of the human gut microbiome. *Sci Rep.* 2023 Jun 24;13(1):10279.

Re: Spectrum02953-24R1 (Skin bacterial community dynamics of hands and forearms before and after military field exercise)

Dear Prof. Trine B Rounge:

Thank you for carefully addressing the Reviewers' comments. I would hereby like to congratulate you on the acceptance of your manuscript for publication in Spectrum!

Your manuscript has been accepted, and I am forwarding it to the ASM production staff for publication. Your paper will first be checked to make sure all elements meet the technical requirements. ASM staff will contact you if anything needs to be revised before copyediting and production can begin. Otherwise, you will be notified when your proofs are ready to be viewed.

Sincerely,
Jan Claesen
Editor
Microbiology Spectrum